# Evolution of Primary Research Studies in Digital Interventions for Mental Well-Being Promotion from 2004 to 2023: A Bibliometric Analysis of Studies on the Web of Science

**DOI:** 10.3390/ijerph21030375

**Published:** 2024-03-21

**Authors:** Maria Armaou, Matthew Pears, Stathis Th. Konstantinidis, Holly Blake

**Affiliations:** 1School of Health Sciences, University of Nottingham, Nottingham NG7 2UH, UK; matthew.pears@nottingham.ac.uk (M.P.); stathis.konstantinidis@nottingham.ac.uk (S.T.K.); holly.blake@nottingham.ac.uk (H.B.); 2NIHR Nottingham Biomedical Research Centre, Nottingham NG7 2UH, UK

**Keywords:** digital interventions, mental well-being, bibliometric analysis, mental health prevention

## Abstract

Research into digital interventions for mental well-being promotion has grown in recent years, fuelled by the need to improve mental health prevention strategies and respond to challenges arising from the coronavirus (COVID-19) pandemic. This bibliometric analysis provides a structured overview of publication trends and themes in primary research studies reporting an array of digital interventions indexed at WoS from 2004 to 2023. Bibliometric data were collected on a sample of 1117 documents and analysed using the Biblioshiny package. Supplemental network visualisation analysis was conducted using VosViewer. The study, based on Web of Science and Scopus databases, indicates a marked increase in publications post-2020. There were seven groups of research themes clustered around “Mindfulness”, “Anxiety”, “COVID-19”, “Acceptance and Commitment Therapy”, “Depression”, “Web-based”, and “Positive Psychology”. Further, results demonstrated the growth of specific themes (e.g., mindfulness, mhealth), the defining impact of COVID-19 studies, and the importance of both randomised controlled trials and formative research. Overall, research in the field is still early in its development and is expected to continue to grow. Findings highlight the field’s dynamic response to societal and technological changes, suggesting a future trajectory that leans increasingly on digital platforms for mental health promotion and intervention. Finally, study limitations and implications for future studies are discussed.

## 1. Introduction

Common mental disorders (CMDs) (e.g., depressive and anxiety disorders) account for a significant portion of the impact of mental disorders on global disease burden [1]. In 2019, depressive and anxiety disorders ranked, respectively, as the 13th and 24th leading causes of poor health worldwide, with 80.6% of the burden occurring among working-age individuals (age 16–65) [2]. In high-income countries, the prevalence rate of mental disorders remains high despite increases in the provision of treatment; this further highlights the importance of implementing quality preventive strategies for the onset of common mental disorders [3].

Preventive interventions entail a large spectrum of interventions across the lifespan that aim to reduce symptoms, reduce individual-level risk factors, and strengthen protective factors to hinder the development and severity of mental disorders [4]. Such interventions are typically classified as universal, selective, or indicated preventive measures for mental illness [4,5]. Universal preventive interventions are offered to the general population and can be beneficial across different population sub-groups. Selective interventions are directed specifically to individuals that have an elevated risk of developing mental disorders, and indicated preventive interventions are specifically directed towards those who have developed minor but observable symptomatology [4]. Mental well-being promotion is advocated as a protective factor that could prevent the onset of CMDs [6,7]. Reviews have highlighted numerous issues associated with the study of preventive interventions across different areas of research focus. For example, mindfulness courses and mindfulness-informed interventions have increased in popularity, especially in high-income countries. However, meta-analytic data demonstrate there is inconclusive evidence regarding their effectiveness within some populations, seemingly being beneficial for some more than others (e.g., students). Due to the studies’ heterogeneity, findings are often not generalisable across different intervention settings [8,9]. Furthermore, cultural and social differences may account for any observed effects. Yet, mindfulness interventions have been shown to be equal in effectiveness to other mental health promotion interventions, and they are consistently more efficacious among those already at elevated risk for mental health problems [8,9,10].

Digital tools and technologies have the potential to improve the scalability and reachability of psychological interventions [11]. Interventions may be either guided (delivered via a trained professional) or self-guided (users working through the intervention on their own). Guided interventions are more frequently used for selective and indicated prevention, whereas self-guided ones are more frequently associated with universal prevention [12,13,14,15]. Self-guided internet- and mobile-based interventions based on common psychotherapeutic techniques can have small positive effects in reducing depression and smaller effects in reducing anxiety [13]. Without distinguishing between interventions that are self-guided and those that are minimally facilitated by expert personnel, moderate-in-magnitude effects on depression and anxiety have been found for internet and mobile-based interventions delivered to adults with chronic physical conditions [16]. Yet, studies are characterised by an overreliance on symptom measures, the limited assessment of comorbidities, and the relatively less frequent assessment of risk reduction and protective factors, which make the distinction between the three types of preventive approaches less clear [13]. Further, when intervention effects have been compared with those of rigorous control conditions, the effects can be weaker [17]. Finally, digital mental health interventions may be a better choice than ‘doing nothing’, but there are persistent uncertainties with regard to intervention effects [18].

Nevertheless, in recent years, enthusiasm for digital interventions aimed at preventing and treating mental health issues has surged significantly, a development propelled by the challenges introduced by the coronavirus (COVID-19) pandemic. Illustratively, the Digital Mental Health section within *Frontiers in Digital Health* published an editorial in 2021/2022 that spotlighted highly engaging articles, emphasising the critical role of digital technologies in mitigating the impacts of the COVID-19 pandemic [19].

The literature, though, surrounding mental health prevention points to the complexity of assessing risk and protective factors for mental health, which is also echoed in discussions regarding the implications of evidence describing mental illness and positive mental health as two interrelated but distinct continua [20]. For this reason, it is imperative to overview the current state of knowledge production in studies reporting digital interventions for mental well-being promotion. This will identify the main drivers of research activity and highlight areas for future research.

The overarching aim of this study is twofold: (1) to comprehensively map the overall landscape of primary research studies in digital interventions for mental well-being, and (2) to identify the most impactful knowledge contributions in this domain. By analysing publication trends and citation patterns, the study sought to outline the evolution and current state of the field. A further aim was to pinpoint influential authors, institutions, and geographical regions that have significantly shaped the trajectory of research in digital interventions targeting mental well-being. For this reason, the study’s objectives were as follows:(1)To identify the most impactful contributors to the knowledge area;(2)To explore influential patterns of research interests that have driven knowledge production;(3)To identify foundational, established, emerging, and niche themes within this knowledge area;(4)To explore shifts in research themes over time.

## 2. Materials and Methods

This bibliometric analysis is reported based on the guidelines on methodological transparency and robust quantitative approaches advocated advocated from a recently developed framework named BIBLIO [21], along with more detailed guidance on scientific mapping [22,23] and the adoption of expansive and inclusive practices for the selection of bibliometric indicators [24]. This enabled us to present our data transparently, facilitating replication and comprehension of the review’s breadth, and ensured the analysis remained responsible and reflective of the multifaceted nature of this research domain.

We employed bibliometric analysis because it offers a structured, quantitative method to synthesise a wide array of publications within the evolving field of digital health interventions. This approach can not only capture the extensive data available but also allows for the identification of research trends, influential works, and key thematic shifts over time. Alternative methods like narrative reviews or expert opinion syntheses, while valuable, do not offer the same level of objective, data-driven insight, particularly in our expansive and multifaceted domain. Hence, bibliometrics was the most fitting approach to provide a broad, unbiased, and transparent overview of the field’s progressions and knowledge networks. Bibliometric analysis provided a transparent, replicable framework that supports the examination of intellectual output and impact, thus making it an ideal tool to disseminate the intellectual landscape of digital mental health research comprehensively.

### 2.1. Search Engines

The bibliometric analysis utilised the Web of Science (WoS) and was complimented by further searches in Scopus. Research has established that, while Scopus has greater breadth in its inclusion of academic journals, WoS is known for its greater keyword sensitivity and a historically extensive volume of references [25,26]. However, differences in the indexing of the articles within each database often mean that the citation patterns differ between them. To bridge these discrepancies, additional manual searches were conducted on WoS for specific titles found on Scopus.

### 2.2. Search Strategy

Searches using combinations of keywords related to ‘digital interventions’ and ‘mental wellbeing’ were conducted in the title, abstract, and keywords sections of the articles for the period from 2004 to 2023. Boolean operators like “AND” and “OR” were used to combine search terms effectively. The timeframe, starting in 2004—the first year with multiple relevant papers in WoS—reflects the rise of digital interventions for mental well-being from the early 2000s. The WoS search string was as follows:

(digital or online or on-line or internet-based or internet* or web-based or web* or computer-based or app* or mobile* or computer* or wearable or virtual or chatbot) (Title) and (psychological well-being or psychological well-being or resilience or stress* or mental well-being or mental well-being) (Topic) and (e-learning or training or stress prevention or positive psychology or stress management or problem solving or problem-solving or self-help or acceptance or self-help or CBT or compassion* or self-compassion or self-compassion or mindfulness or mindfulness-based or cognitive behav* therapy) (Abstract) and Article or Proceeding Paper or Early Access or Meeting Abstract (Document Types) and Book Chapters or Review Article or Editorial Material or Retracted Publication or Book Review or Correction or Letter or Item Withdrawal or Retraction or Withdrawn Publication or Data Paper (Exclude—Document Types) and Chapter (Exclude—Search within all fields) and Protocol (Exclude—Search within all fields) and Meta-analysis (Exclude—Search within all fields) and Systematic Review (Exclude—Search within all fields) and Disorder (Exclude—Search within all fields)

The Scopus search string was as follows:

(TITLE (digital OR e-learning OR online OR on-line OR internet-based OR internet* OR web-based OR web* OR computer-based OR app* OR mobile* OR computer* OR wearable OR virtual OR chatbot) AND TITLE-ABS-KEY (psychological AND well-being OR resilience OR stress* OR mental AND well-being OR mental AND health) AND NOT TITLE-ABS-KEY (meta-analysis OR systematic AND review OR scoping AND review) AND NOT TITLE-ABS-KEY (disorders OR disorder) AND NOT TITLE-ABS-KEY (protocol)) AND PUBYEAR > 2003 AND (LIMIT-TO (DOCTYPE, “ar”) OR LIMIT-TO (DOCTYPE, “cp”))

### 2.3. Eligibility Criteria

Titles and abstracts were screened applying the following inclusion criteria: study population (any), study design (any), intervention (digital psychological or educational interventions), outcomes (psychological/psychosocial outcomes). Included papers could be as follows: Article or Proceeding Paper or Early Access or Meeting Abstract. Papers were excluded if they were any of the following: Book Chapters or Review or Editorial, Retracted Publication or Book Review or Correction or Item Withdrawal or Retraction or Withdrawn Publication or Data Paper. Documents were excluded if they reported interventions focused on the treatment of psychiatric disorders or did not target specific psychological or psychosocial outcomes. No restrictions to language were applied. Review papers, meta-analyses, protocols, and theoretical pieces were excluded. In cases where it was unclear from the abstract if the intervention was digitally delivered, the full paper was then retrieved.

### 2.4. Data Selection Procedure and Dataset Refinement

Documents that matched the inclusion and exclusion criteria were marked on the researcher’s WoS and Scopus accounts (Figure 1). A total of 11,388 documents were identified in the WoS and 4384 in Scopus, published between 2004 and 2023. Based on the study inclusion and exclusion criteria, 392 eligible documents were identified in Scopus and, initially, about more than 900 documents were identified at the WoS Core Collection. Subsequently, manual searches were conducted at WoS to retrieve the documents already identified on Scopus. In the WoS Core library, 89% Scopus documents (n = 349) were identified. Twenty-one additional duplicate entries were identified following manual searches using the document’s title or author’s name (previously retrieved by Scopus). On certain occasions (n = 7), documents were identified after searches to all the databases in the WoS, and nineteen documents were not indexed at the WoS databases at all. Finally, 1131 documents were generally identified through the WoS, of which 1124 belonged to the WoS Core Collection and were downloaded for further analysis.

### 2.5. Analysis

The final dataset used for this study was downloaded directly from the WoS in .txt format. Two independent researchers performed the data analysis, subsequently engaging in iterative discussions to reconcile discrepancies and achieve consensus on the findings. This collaborative approach ensured a reliable and accurate representation of the data. The analysis was conducted using Biblioshiny 4.1, available within the Bibliometrix package in RStudio 1 [27]. This open-source software provides a suite of statistical tools and functions for bibliometric analysis. It facilitates detailed productivity analysis, including publication and citation rates, as well as identifying key contributors to the field. Its tools can give an assessment of the h-index, m-index, and g-index, which are bibliometric indicators used to evaluate a researcher’s publication impact and productivity. The h-index reflects the number of publications (h) with at least h citations each, balancing productivity with citation impact. The m-index, calculated by dividing the h-index by the number of years since the first publication, adjusts for career length, offering insight into sustained impact. The g-index addresses the h-index’s limitation by emphasising highly cited papers, defined as the largest number (g) where the top g articles have collectively received at least g^2^ citations. Interpretation of these indices varies across disciplines, with low, moderate, and high benchmarks generally considered as follows: an h-index of 0–10, 10–20, and 20+, respectively; an m-index below 1, between 1–2, and above 2; and a g-index that closely follows, significantly exceeds, or doubles the h-index for low, moderate, and high categories.

Moreover, it supports comprehensive science mapping through citation analysis, keyword co-occurrence analysis, and co-authorship analysis, enabling a deeper understanding of the research landscape. Combined with network analysis, science mapping can illustrate the bibliometric and intellectual structure of different areas of research focus [22,23]. Additional network maps were produced on VosViewer (V. 1.6.19) to explore the characteristics of emergent themes.

## 3. Results

### 3.1. Output of General Information and Annual Information

Source .txt data were uploaded on Biblioshiny 4.1. and following the review of their general information, 1117 documents were included in the final analysis. Seven records were excluded for the purpose of generating the quantitative analysis in accordance with the study’s inclusion criteria. The records excluded were either reported in a letter, an editorial, or a book chapter, or they were counted as being published in 2024. Descriptive information regarding the overall dataset included the following: DATA INFORMATION (Journals: 489, Documents: 1117, Document Average Age: 4.47, Average citations per doc 16.44); DOCUMENT CONTENTS (Keywords Plus: 1575, Author’s Keywords: 2189); AUTHORS (Authors: 4969, Authors of single-authored documents: 34); AUTHORS COLLABORATION (Single-authored docs: 34, Co-Authors per doc: 5.46, International co-authorships: 20.23%); DOCUMENT TYPES (articles: 948, article early access: 37, article; were proceedings papers: 5, meeting abstracts: 58, proceedings papers: 69).

### 3.2. Performance Analysis

Figure 2 shows that publication rates increased after 2012, with 2014 being the first year that had more than 20 publications in the field. Subsequently, publication rates consistently rose until 2020. A significant surge in publications occurred from 2020 to 2022, being the first possible indicator of the effect of the COVID-19 pandemic on this research landscape. Further, 52% of all publications were generated between 2020 and 2023. The figure also highlights variations in citation rates, identifying peaks in 2005 and 2013, yet a decrease in citations as the annual production increases.

#### 3.2.1. Authors

Table 1 shows the authors who had the highest number of publications in the field. Overall, most of the documents were written in English by authors affiliated with institutions in high-income countries. R. Lappalainen emerges as the most prolific, authoring 17 articles with a collaborative score indicating a significant partnership in research. P. Lappalainen follows with 15 articles, demonstrating similar networking. D. Lehr and D.D. Ebert contribute likewise with 15 and 14 articles, respectively. T. Berger and M.E. Levin each have 13 articles to their names, with Levin’s higher collaborative score suggesting more extensive co-authorship. K. Kaipainen, with 11 articles, alongside G. Andersson, H. Christensen, and P. Cuijpers, each contributing 10 articles, display active and still high engagement. Total citation amounts are similar for all but T. Bergen and K. Kaipeainen, with less than half TC counts.

##### Authors’ Production over Time

In 2016, D. Lehr and D.D. Ebert each published two articles, garnering 189 citations, averaging an annual impact of 21 citations. That same year, P. Lappalainen and R. Lappalainen also contributed two articles, achieving 164 citations, at an average of 18.22 citations per year. M.E. Levin’s 2017 publication received 125 citations, translating to 15.63 citations annually. Fast forward to 2018, D. Lehr’s work saw a total of 170 citations across four articles, while D.D. Ebert’s three articles earned 166 citations, and P. Cuijpers’s single work received 115 citations. In 2020, H. Christensen’s research secured 66 citations, averaging 13.2 per year. Yet trends started to change as, throughout 2021 and into 2022, R. Lappalainen published on ACT, with articles cited up to 11 times, paralleled by D. Lehr and D.D. Ebert, who expanded on web-based stress interventions and depression prevention. By 2023, the focus shifted towards adolescent mental health amidst COVID-19, with R. Lappalainen’s article receiving five citations, indicating evolving research directions and emerging impacts.

Figure 3 shows the bibliographical coupling of authors with more than five documents and more than five citations within our sample. This refers to the intellectual similarity between authors as manifested by their citations in other documents. The red cluster consists of nine authors whose work is mostly associated with digitally delivered mindfulness interventions. The green cluster also includes nine authors with outputs focusing on digital mental health interventions, often based on cognitive behavioural therapy. The blue cluster of authors includes eight authors with outputs focusing on interventions based on acceptance and commitment therapy. The yellow cluster includes seven authors whose work focuses on web-based interventions, often for depression and stress reduction. Finally, the purple cluster includes seven authors whose work focuses on digitally delivered interventions for mental health prevention and treatment.

##### Most Local Cited Authors

This analysis revealed the local citation impact of several authors within their research community. R. Lappalainen leads with 90 citations, indicating the strongest influence in the field. Close behind, M.E. Levin has collected 78 citations. K. Cavanagh follows with 75 citations. C. Strauss, with 72 citations, and P. Lappalainen, with 70 citations, both exhibit noteworthy influence. As demonstrated in the network analysis in the subsequent sections. F. Jones, with 63 citations, D. Lehr, with 62 citations, and H. Riper, with 61 citations, each have made impactful contributions. D.D. Ebert and M. Berking conclude the list with 60 and 58 citations, respectively, underscoring their roles as influential figures in their research community. There was no significant gap between authors in terms of their citation counts, highlighting a competitive and collaborative environment.

##### Most Relevant Sources

The productivity of journals was ranked using the Bradford law, which ranks documents in zones of productivity, where the first zone includes a core group of the most productive journals (Table 2). The bibliometric analysis identified the *Journal of Medical Internet Research* as the top source for articles, contributing 62 articles to the dataset. Following closely, *JMIR Formative Research* and *JMIR Mental Health* supply 50 and 43 articles, respectively, indicating a strong focus on digital health research. *Mindfulness* and *Frontiers in Psychology* contribute 37 and 36 articles, respectively, pointing to an interest in psychological and mindfulness-based interventions. *JMIR mHealth and uHealth* adds 28 articles, highlighting the growing importance of mobile health technologies. The *International Journal of Environmental Research and Public Health* provides 27 articles. *Internet Interventions-The Application of Information Technology in Mental and Behavioural Health* contributed 23 articles, highlighting the role of digital interventions in mental health. *Annals of Behavioral Medicine* and *Psycho-Oncology* offer 18 and 17 articles, respectively. The Journal of Medical Internet Research had the highest total citation number of 1864, followed by *JMIR mHealth and uHealth* with 1054. This distribution of sources and articles showcases a broad spectrum of research interests within the field, ranging from digital health innovations to environmental health research and psychological interventions. The *JMIR Formative Research* is a journal that showed the most recent growth with relevant publications beginning in 2020.

##### Sources Local Impact

The bibliometric indicators for a selection of journals reveal significant insights into their impact and contributions to their respective fields. The *Journal of Medical Internet Research* stands out with an h-index of 22 and a g-index of 42, reflecting a strong influence within the medical internet research community, as demonstrated by a total citation (TC) count of 1864 from 62 published articles since 2005. Its m-index of 1.10 suggests a consistent output of impactful research over time. *JMIR Mental Health*, with an h-index of 20 and a g-index of 29, shows a notable impact in mental health research, particularly since its inception in 2016. It has accrued 891 citations from 43 articles, with a high m-index of 2.222, indicating rapid recognition in its field. *JMIR mHealth and uHealth* presented an h-index of 17 and a g-index of 28, evidence of its significance in mobile health research. Since 2013, it has garnered 1054 citations from 28 articles, with an m-index of 1.417, underscoring its growing influence in health technology. *Mindfulness*, with an h-index of 15 and a g-index of 26, has amassed 701 citations from 37 articles since 2014. Its m-index of 1.36 indicates steady academic contribution and influence. *Internet Interventions-The Application of Information Technology in Mental and Behavioural Health*, although newer with a start year of 2017, shows a promising start with an h-index of 11 and a g-index of 15, totalling 244 citations from 23 articles. Its m-index of 1.37 suggests a rapidly establishing presence in its niche. *Frontiers in Psychology*, with an h-index of 10 and a g-index of 19 since 2016, has accumulated 401 citations from 36 articles. Its m-index of 1.11 indicates solid performance in the psychology field, reflecting its role in advancing psychological research.

As Figure 4 shows most of the documents were published in open-access journals. Among the top 13 most relevant sources, only 5 journals (*Mindfulness*, *Annals of Behavioral Medicine*, *Psycho-oncology*, *Journal of Contextual Behavioral Science*, and *Behaviour Research and Therapy*) offer both an open-access and subscription options. Publication fees for open-access publications in the top 13 most relevant sources are over USD 2000 ranging up to USD 4940.

#### 3.2.2. Most Relevant Affiliations

The analysis identifies the top academic institutions by their contribution to the dataset, with Harvard University, the University of New South Wales Sydney, and Vrije Universiteit Amsterdam leading, each contributing 37 articles. Close behind are the University of London and the University of Sydney, each with 36 articles, followed by the University of Melbourne with 34 articles, and the University of California System with 33 articles. Seoul National University (SNU) contributed 31 articles, while University College London provided 27 articles. Monash University and Karolinska Institutet contributed 22 and 21 articles, respectively. All other affiliations, beyond those highlighted, contributed to 20 or fewer articles, indicating a concentration of research output within these leading institutions.

##### Affiliations’ Production over Time

The production of articles over time by leading affiliations demonstrates a dynamic and growing contribution to scholarly work. Vrije Universiteit Amsterdam, the University of New South Wales Sydney, and Harvard University exhibit a consistent increase in their output, each peaking at 37 articles by 2023. The University of Sydney and the University of London also made significant contributions, with both institutions reporting 36 articles by 2023. The data reveals a progressive increase in the University of Sydney’s output from 22 articles in 2018 to 36 articles by 2023, illustrating a 63% upward trend in research productivity. Vrije Universiteit Amsterdam’s output has risen 37% from 27 articles in both 2018 and 2019, reaching 37 articles by 2023 and 2024. Similarly, the University of New South Wales Sydney and Harvard University have shown a marked increase in their outputs, moving from lower figures in earlier years to reaching their peak by 2023. This upward trajectory in article production across these prestigious institutions not only underscores their significant role in advancing research but also highlights the increasing momentum of academic contributions over time, with each institution demonstrating a commitment to expanding their research footprint and impact in their respective fields.

##### Scientific Production across Countries

The cumulative number of article contributions for each country, based on the data processed, is as follows:

The bibliometric data presents a comprehensive overview of 17,632 articles produced over the studied time frame, with the United States leading the count at 5237 articles (30%). The United Kingdom follows with a substantial output of 3345 articles (19%), while Australia ranks third with 2225 articles (12%). The Netherlands and Germany also show high productivity, with 1323 (7.5%) and 924 articles (5%), respectively. With less than 5% contributions, Sweden contributes 737 articles (4%), evidencing robust research activity, and Canada’s contribution of 641 (3%) articles underscores its role in the academic landscape. China’s scholarly output is noted at 613 articles, while Finland demonstrates considerable research contributions with 546 articles. Switzerland and Japan have produced 387 and 348 articles, respectively, indicating their active participation in research. Ireland, with 312 articles, Spain with 257, and Italy with 231, also feature prominently in the dataset. New Zealand’s output stands at 206 articles, followed by Korea with 189 and India with 111, reflecting the global spread of research contributions in the field. These figures highlight not only the volume of research emanating from these countries but also uncover that the breadth of international engagement in academic discourse is narrower when considering production over location.

##### Most Cited Countries

In terms of citations, the United States again stands out with a total of 5237 citations, averaging 16.30 citations per article, indicating the broad impact of its research. The United Kingdom follows, with average article citations of 30.10, reflecting the high quality and influence of its research outputs. Australia’s contributions are also notable, with 2225 citations at an average of 22.50 citations per article, showcasing the significant impact of its scientific work. The Netherlands exhibits the strongest metric as well, with an average of 25.00, highlighting the influence of their research contributions. Germany, Sweden, Canada, China, and Finland also show similar significant impacts.

##### Collaboration Network

The analysis of international research collaborations highlights significant partnerships across the globe. Germany and the Netherlands lead with 14 collaborations, evidencing strong European research ties. Equally notable are the connections between Australia and the United Kingdom, and between the USA and both Canada and the United Kingdom, each registering 13 collaborations. The USA’s collaboration with China (12 instances) underscores its global research connectivity. Additional key partnerships include the USA’s collaborations with Australia, the Netherlands, and Norway, and the regional Nordic cooperation between Sweden and Finland, each showcasing the USA’s central role and the importance of regional and transatlantic partnerships in fostering a globally interconnected research community. Figure 5 shows a collaboration map of the affiliations of the correspondent authors. Harvard University was the only USA-based institution with collaborations with both European and Asian universities. The collaboration map demonstrates that many collaborations were defined by the geographical proximity of institutions, such as collaborations between European institutions or between Australian institutions.

#### 3.2.3. Most Globally Cited Documents

Table 3 presents the top 10 highly cited research papers, spotlighting their influence through total citations, yearly citation rates, and adjusted citation impacts. Leading this distinguished group, an article by Blake H. in the *International Journal of Environmental Research and Public Health* from 2020 stands out for its notable citation metrics, illustrating its substantial global impact shortly after publication. Other significant contributions include works by Gilbody S. in the *BMJ: British Medical Journal*, Cavanagh K. in *Behaviour Research and Therapy*, Clarke G. in the *Journal of Medical Internet Research*, and Inkster B. in *JMIR mHealth and uHealth*, each demonstrating considerable influence through their citation figures and rates. Additional important studies from journals like the *Journal of Happiness Studies*, *JAMA Psychiatry*, and the *Journal of Occupational Health Psychology* further exemplify the breadth of impactful research across various domains.

#### 3.2.4. Most Locally Cited References

The dataset highlights foundational references within the research domain, marked by their significant citation counts. Cohen et al’s 1983 study on health and social behaviour [38] leads with 174 citations, emphasising its seminal role. Cohen J.’s 1988 work [39] on statistical power analysis in behavioural sciences follows with 108 citations, showcasing its critical methodological influence. Articles by Brown K.W. et al. and Kabat-Zinn J. from 2003 [40,41], each focusing on mindfulness and its psychological applications, gather 91 and 85 citations, respectively, reflecting their impact. Spijkerman M.P.J. et al. 2016 clinical review [42] and works by Braun V. and Kroenke K. from 2006 [43,44], each earning 75 citations, contribute significantly to psychological interventions and symptom assessment. Publications by Spitzer R.L. in 2006 [45] and Christensen H. in 2009 [46], with 69 and 66 citations, highlight the importance of internet-based mental health interventions. Baer R.A.’s 2006 [47] article on mindfulness-based practice assessment has 64 citations, underscoring its relevance. These references serve as cornerstones in health, psychology, and behavioural sciences, demonstrating their foundational influence and enduring relevance.

### 3.3. Concept Mapping

Assessing authors’ keyword frequency can describe the main research themes in the field. A co-occurrence network of the top 250 authors’ keywords demonstrated seven main research themes (Table 4). Cluster 1 was the theme with the most frequent keywords of mindfulness (n = 220), mental health (n = 168), and stress (n = 146). The second cluster had the largest number of keywords, including anxiety (n = 90) and mhealth (n = 58). The keyword with the highest frequency for cluster three was ‘COVID-19’ (n = 102), for cluster four was ‘acceptance and commitment therapy’ (n = 61), and for cluster five, it was ‘depression’ (n = 126). The sixth cluster had two main keywords, ‘web-based’ (n = 24) and ‘college student’ (n = 19), and cluster seven had one main keyword, ‘positive psychology’ (n = 19).

An analysis of the bibliometric features of keyword clusters can provide insights into the development and cohesion of different areas of research activity. Each research theme can be described in terms of importance (or centrality) and density. Its centrality is defined by the extent of its relationships with other research themes/keywords, and its density refers to the total number of documents that include the same keyword [48]. A network analysis can examine those key features of different research themes by placing them within four quadrants that refer to a differing degree of development of research themes [49]:Upper-right quadrant: core or mainstream themes; the motor themes, which are both important and well-developed. They have maximum density and centrality and represent a large portion of the research during a specific time-period.Lower-right quadrant: developed but isolated themes; basic themes that are important but not yet well developed. They demonstrate centrality, but they are not yet mature, although they have the potential to grow.Upper-left quadrant: established niches of research. They exhibit low centrality, and they appear separate from the overall focus of research in the field.Lower-left quadrant: emerging or declining themes. They are themes of low importance and low development that are either emerging or declining themes within a specific research field.

A thematic mapping of the studies’ research themes was completed to explore their degree of development (Figure 6).

#### 3.3.1. Motor Themes

Two clusters of themes are shown to be pertinent to the research development of the field. The most well-developed one involves research in depression, anxiety, and mhealth, followed by a smaller cluster of themes focusing on quality of life, cancer, positive psychology, and psychological distress. Finally, the research cluster with a focus on adolescents, randomised controlled trial designs, psychological flexibility, and adherence placed them near the centre of the two axes, standing right between niche and motor themes.

#### 3.3.2. Basic Themes

The themes of mindfulness, mental health, stress, and COVID-19 have the highest level of density in the sample, which means that they consistently co-occurred as keywords in other documents. However, they still lack development as they are low in centrality, which means that they exhibit little relationship with other clusters of themes within the thematic network. A cluster whose content appears to be closer to becoming a motor theme involves research focused on acceptance and commitment therapy, online intervention, self-help, and university students. Finally, the lowest in centrality but with overall good degree of density was the cluster of themes focusing on stress management, virtual reality, and mindfulness meditation.

#### 3.3.3. Niche Themes

There were two clusters of themes that represented niche themes of research activity. One cluster included the theme of usability, and the other included the themes of engagement and health promotion. Both of those clusters appeared to be close to one another and exhibited low density but good centrality. This means that they had little coverage overall across the documents but were associated with other thematic clusters.

#### 3.3.4. Emerging or Declining Themes

There were also two clusters of themes representing emerging or declining areas of research focus, and those were e-health and chronic pain. Among those, chronic pain was the one closest to the centre of the two axes. Finally, research associated with web-based interventions had a relatively low density but was right between the two quadrants of niche and declining or emerging themes. Their development status and interrelationships with other themes were further explored by the network analysis produced on VosViewer, which will be presented further below.

### 3.4. Concept Evolution

The period of review, 2004–2023, was divided into three sub-periods: 2004–2015, 2016–2020, and 2021–2023. This way, a separate bibliometric strategic map was developed for each of these three periods. These cut-offs were selected as they match the time-points at which the most noticeable changes in the publication rates were observed (Figure 1). The first period (2004–2015) is the longest one because a sufficient volume of publications is required to complete the identification of the structural characteristics of a research field and map their subsequent evolution over time. The second and third periods (2016–2020, 2021–2023) are shorter, as they are mostly focusing on emerging or declining trends and areas of future research growth. Overall, 134 documents were analysed for the first period, 402 for the second period, and 581 for the third one. The output from RStudio’s Biblioshiny is used to portray the concept evolution for each of the time periods (Figure 5).

Figure 7 demonstrates the considerable increase in research focus on mindfulness in recent years. Furthermore, it illustrates the growth of mHealth as a dominant research theme and the central role that studies associated with the COVID-19 pandemic played in the development of the research. Finally, acceptance and commitment therapy appears to be consistently the most relevant intervention approach, even following the shifts that occurred in research development following the emergence of COVID-19 studies.

#### 3.4.1. Period 2004–2015

Figure 8 shows five clusters of motor themes. The themes with the highest density were research on depression and internet interventions. Furthermore, cognitive-behavioural therapy and e-learning were significant motor themes that exhibited slightly higher centrality and lower density. Finally, in the cluster of themes focusing on randomised controlled trial designs and cost effectiveness, the motor theme had the least density but was still significantly important for overall research development. During this time, acceptance and commitment therapy was a basic theme that was not well developed and exhibited low centrality. Mindfulness, along with resilience, was almost at the centre of the two axes but tilted towards the basic themes’ quadrant. At the same time, the cluster involving references to family caregivers and computers was the smallest one with the lowest levels of centrality and density research themes, while the other cluster, classified as emergent or in decline, grew in maturity in later years.

#### 3.4.2. Period 2016–2020

Figure 9 shows two clearly well-established themes during this period. Research focusing on burnout and mobile applications and college/university students signify the most well-developed areas of enquiry. Those were followed by research focusing on ehealth and cancer and cognitive behavioural therapy. For the first time, a psychological outcome such as burnout and a reference to mhealth (i.e., mobile applications) are the motor themes with the highest centrality. The larger cluster of themes that were most frequently mentioned in other studies’ keywords involved were ehealth and cancer, although not fully reaching motor-theme status. This is a research period characterised by research associated with the COVID-19 pandemic quickly at the centre of research importance, followed by psychoeducation and web-based intervention. Mindfulness and well-being, which have been among the most widely reported areas of research, appeared to still be not well-developed, while virtual reality appeared to be a rapidly growing research area. On the other hand, positive psychology interventions and psychological well-being were classified as niche themes along with internet-based interventions. At the same time, psychological stress, psychological distress, and online intervention were classified as emerging or declining themes.

#### 3.4.3. Period 2021–2023

This period spots two well-established cluster themes, namely e-health and chronic pain, as well as mhealth and mobile, with the first two demonstrating higher centrality. Mhealth/mobile phones represent the most frequent area of study (Figure 10). Similarly to the previous period, mindfulness research had the highest density but was less central to the overall research focus. Acceptance and commitment therapy appeared to have increased in maturity, although it was not fully developed as a motor theme. Other areas that are expected to continue to grow are research on the adolescent population, qualitative research, and loneliness. Two new niche themes during that period were technology and biofeedback, with cognitive behavioural therapy also being classified as a niche theme. Research in mental well-being and stress reduction was classified as an emerging research theme based on the growth in articles during this time, which further signals a shift in research interests.

### 3.5. Emergent Themes’ Characteristics

The VosViewer output of the authors’ keyword network analysis was to examine the characteristics of emergent themes and indicate the ones that appear to be in decline. The network analysis (Figure 11) shows the associations between themes that were classified as declining or emergent, along with their average publication years. It is evident that e-health, chronic pain, web-based interventions, meditation, and virtual reality are research themes that have continued to grow. On the other hand, “computer” and “family caregivers”, which were the earliest occurring research themes with an average publication year of 2015.00 and 2018.50, respectively, did not follow the same development trajectory.

“Psychological distress” (Avg. pub. year: 2021.06) and “mental wellbeing (Avg. pub. year: 2021.60) were the most recently emerged targeted outcomes of digital interventions and demonstrate multiple links with other emergent themes. At the same time, there appears to be a shift in relevant targeted outcomes of digital interventions that is reflected in “psychological stress” (Avg. pub. year: 2018.31) having the fewest links with the other keywords in this group.

Finally, Figure 12a–f show the associations that were specifically relevant to quality of life (Avg. pub. year: 2019.78), psychological distress (Avg. pub. year: 2021.06), loneliness (Avg. pub. year: 2021.67), mental wellbeing (Avg. pub. year: 2021.60), psychological wellbeing 2020.24, and wellbeing (Avg. pub. year: 2020.63). Except for quality of life, which was already a well-developed theme before the beginning of the pandemic, the research development of the rest of the themes seems to be interlinked with the research production triggered by the COVID-19 pandemic.

## 4. Discussion

The objective of this study was to examine the international contours of primary studies reporting on digital interventions for mental well-being promotion as indexed in the WoS. Bibliometric data were collated from the WoS for studies published between 2004 and 2023. Bibliometrix and Biblioshiny were used to describe in detail the main knowledge contributors in the field and identify the key trends that have shaped the literature in the field and the ways in which research themes have evolved. The visual aids of Vosviewer mapped the characteristics of emerging research themes.

### 4.1. Research Contributors’ Impact

The key knowledge contributors were assessed in terms of productivity rates and impact, citation rates, and collaboration patterns. The publication rate of studies has increasingly progressed since 2004, with some of the most productive authors beginning to contribute to the field between 2013 and 2015. A distinctive increase in research productivity, though, was observed early on in 2020 with the beginning of the COVID-19 pandemic, which was followed by a surge in publications reporting on preventive digital mental health interventions, reflecting the fast-track submission processes that were adopted by academic journals to speed up knowledge dissemination. The *Journal of Medical Internet Research* was the leading and most influential source of publications, which also ranked first in a previous bibliometric analysis of research production in technology and psychotherapeutic interventions [50]. *JMIR Formative Research* ranked second, further highlighting the surge in interest in developing and, thus, piloting and evaluating novel interventions during the last three years. Overall, the USA, Australia, and the UK had the largest number of studies, while the affiliations with the largest number of contributions were the Vrije Universiteit Amsterdam, the University of New South Wales Sydney, and Harvard University. A closer examination of the collaboration networks between affiliations across countries also revealed how knowledge production was mostly driven by a combination of <10 international partnerships and strong regional ones. Overall, most of the work focused on was produced by authors with institution affiliations in high-income countries with open-access publications driving research publications. This means that the studies’ insights may be less relevant to resource-poor settings, especially low- and middle-income countries. The key research interests of the most relevant authors included digitally delivered mindfulness-based interventions, interventions based on acceptance and commitment therapy, and digitally delivered interventions for mental health prevention.

### 4.2. Influential Patterns of Research: Concepts’ Development and Shifts

The diagrams of concept mapping and the concept evolution of themes as described by the authors’ keywords illustrated which aspects of intervention research have shaped research progress, highlighting notable challenges that have also occurred over time. Some of those themes have consistently driven research activity, while the centrality and the density of other themes vary. The only themes relevant to specific research designs involved randomised controlled trials, which was also among the most important keywords, and qualitative research that gained more traction within the last two years. The centrality of RCTs is an expected finding, as most of the included studies reported on the effectiveness of the interventions. At the same time, references to qualitative research within a cluster of more recent publications that also included loneliness highlight the growing impact of research themes that were developed through studies that were conducted during the COVID-19 pandemic. Furthermore, it coincides with niche themes such as engagement and usability (also associated with feasibility and acceptability) and their associations with a broader range of research designs (e.g., participatory design), which were all in all less frequently represented in the data. This diversification in research methodologies reflects a broader shift towards a more holistic understanding of mental health interventions, recognising the importance of both quantitative outcomes and qualitative insights to address complex psychological phenomena exacerbated by global crises like the COVID-19 pandemic [51,52,53,54].

Specific population groups or contexts included adolescent populations and college students, the workplace, and individuals affected by chronic pain. Such findings are in accordance with the extensive literature reporting on interventions delivered to such participant groups as universal, selective, or indicated mental prevention measures [55,56,57,58,59,60,61,62,63]. Research in the context of COVID-19 was a central driver of research activity. In our sample, research associated with the COVID-19 pandemic was the fifth most frequent research theme. As a result, the cluster of themes about web-based intervention, psychoeducation, and COVID-19 appeared to be the central driver of research development for the whole period 2016–2020, and their positioning demonstrated high importance and continued growth. That was also indicated by the fact that the most highly cited paper within our sample described (and provided links to) the development and evaluation of an open-access digital package providing evidence-based guidance and support for the psychological well-being of healthcare employees, which was widely accessed across the world (over 77,800 users within the first pandemic year) [28]. Epidemiologic data have shown that the COVID-19 pandemic exacerbated the prevalence and burden of depression and anxiety, with geographical variations in impact [64]. Between 2020 and 2021, several reviews were written to inform the implementation of remote solutions for both the treatment and prevention of mental health conditions [65,66]. This research activity was followed by a surge in digital interventions driven by the need to reach population groups whose mental health was at risk the most using remote delivery methods, which had value during regional and national lockdowns and through periods of social distancing measures [67,68,69].

The scientific mapping of the themes produced based on the authors’ keywords produced seven clusters of themes, each referring to at least one of the following intervention characteristics: (a) prevention outcomes, (b) intervention delivery methods, (c) target populations, and/or (d) intervention approaches. The first two clusters were the largest clusters of themes that dominated research production. The first cluster included the most frequently studied psychological outcomes: mental health, stress, well-being, mindfulness, and self-compassion. Within this cluster, the keywords describing interventions’ approaches were “self-help”, “stress management”, and “meditation training”. Intervention delivery within this cluster was most frequently described as online and, in some cases, via virtual reality. References to the target population were “university students” and “adolescents”. The results also showed that although those themes are expected to continue to grow, they are also characterised by low centrality, and they do not appear to have yet become motor themes for research development in the field. A recent umbrella review on the effectiveness of digital mental health interventions for university students also determined that, although web-based interventions and online skills training are at least partially effective in reducing anxiety, stress, and depression, the evidence is inconclusive for the effectiveness of interventions using virtual interventions and relaxation [70]. Such findings highlight the need to establish more systematic approaches to the evaluation of digital mental health interventions [70]. Moreover, mindfulness, which was the most researched theme in our sample, continued to grow after the beginning of the COVID-19 pandemic, but it is still not a well-developed theme.

This occurrence has been frequently explored in reviews examining the effectiveness of digital mindfulness-based interventions and includes implementation obstacles, low quality of evidence, and the emergence of evidence supporting non-digital mindfulness-based interventions as more effective than their digitally delivered counterparts. This highlights a pivotal challenge for digital mental health research: balancing the rapid development and deployment of digital interventions with the rigorous, long-term evaluation necessary to validate their effectiveness and optimise their design for diverse populations [71,72,73].

The second thematic cluster included a larger number of keywords, which, in their majority, were terms referring to technology-enabled interventions. The main characteristic of this group of themes was its multiple references to mobile-delivered interventions. It also included anxiety, which was shown to be among the motor themes of research, and psychological distress, which was more frequently reported in more recent publications. Furthermore, the intervention approach within this cluster was cognitive behavioural therapy, while “workplace” was included as a specific intervention context. Previous reviews have shown that app-delivered interventions that are based on behaviour change strategies across diverse populations, especially those developed for specific populations, can be effective in reducing depression symptoms, anxiety, and stress levels [74].

The third cluster was dominated by studies focusing on the COVID-19 pandemic, with prevention measures focusing on resilience, burnout, well-being, and psychological well-being. Those themes appear to be still early in their development, but they have a great potential for growth, possibly fuelled by a greater variation in discussions and approaches to mental health prevention driven by the COVID-19 impact and the challenges posed for sustainable healthcare in the future [75,76,77].

The fourth cluster features references to online interventions, and its dominant keyword was acceptance and commitment therapy, followed by references to cancer and chronic pain research. Furthermore, it is a cluster that is also defined by randomised controlled trial designs, which highlights the centrality of research studies focusing on acceptance and commitment therapy [61,78,79].

The fifth cluster focused on prevention measures for depression and quality of life, with references to e-health and qualitative research. Overall, depression was among the most frequently reported outcomes, but over time, its centrality as a targeted outcome of preventive interventions appeared to diminish. One reason may be that as research in digital interventions matured, digital interventions focusing on depression began to examine explicitly their effects in the clinical treatment of depression [80,81]. At the same time, a broader range of outcomes (e.g., empirical domains of good mental health) have become more frequently discussed as suitable targets for mental health prevention strategies [6].

The other two clusters included frequently identified research themes (web-based delivery and college students; and positive psychology), with their overall influence on research development more evident through their associations with other research themes (e.g., well-being, self-help).

### 4.3. Limitations

The study had several limitations that need to be acknowledged. First, the final article sampling consisted of articles included only in the WoS database. This means that citations indexed in other databases are not included in this study’s productivity analysis. The advantage of this approach was that the analysis was completed with consistent bibliometric data across one dataset, but as it also became clear during the process of data searches and refinement, there is a small percentage of studies in Scopus that are not indexed in the WoS Core Collection. This may not have had a significant impact on the overall observations regarding the evolution of the main drivers of research activity, but it may have reduced some insights on recent niche concepts representing research work presented in recent conferences. Furthermore, it was also observed that for articles where there was a title available both in English and another language, Scopus indexed both, whereas WoS indexed only the English title or did not index the title or the source at all. Such observations may potentially lead to overall counts of bibliometric data being slightly skewed towards English-speaking entries. An associated limitation of this study was that the sample of documents was dominated by studies conducted in high-income countries, despite the absence of language restrictions in our searches. Furthermore, discrepancies in the data due to missing values, different spellings of the authors’ full names, or differences in affiliations’ names within the source data can all lead to variations in the results of the performance analysis [26]. Moreover, the exclusion of reviews, meta-analyses, and editorial pieces means their contribution to the overall evaluation of knowledge production. Another limitation that generally applies to bibliometric analyses is that research themes are defined by the choice of authors’ keywords, not the actual content of a study; thus, only citations provide an estimate of the importance of specific studies within a field. At the same time, the different spellings of keywords referring to the same concept (e.g., ehealth, e-health, mobile app, apps, etc.) mean that the full scope of a theme becomes easily fragmented. Finally, the concept mapping and evolution diagrams on Biblioshiny were based on the analysis of the clusters for the 250 most frequently used authors’ keywords with a minimum of 10 words per cluster. That approach reduced the nuanced description of the concepts but allowed for a comprehensive depiction of the themes and their key changes across the set time-points.

### 4.4. Implications for Future Research

This study provided a comprehensive overview of the research trends in digital mental well-being promotion. Future research will need to explore how some trends may have had more impact than others in mental health prevention/promotion research. Future systematic reviews and empirical studies should aim to more frequently report implementation outcomes, especially in relation to mindfulness interventions. Future evidence reviews could aim to map the progress of niche areas of research using data from the Scopus dataset and, if required, grey literature to explore research development internationally, aiming to include and evaluate the content of knowledge contributors in languages other than English. Future research will need to specifically target reviewing research outputs produced by authors in low- and middle-income countries, which would require the inclusion of searches in Google Scholar and grey literature. What is more, scoping review techniques can be implemented to conduct searches focusing on specific criteria (e.g., socio-demographic characteristics, health conditions, and contexts). Such an approach can allow a better understanding of the impact of the observed research trends on digital interventions. Finally, future studies should aim to evaluate the accumulation of evidence related to mental health prevention since the emergence of the COVID-19 pandemic as a starting point, considering both its long-term impact and the intersections of that impact with chronic conditions [82,83,84].

## 5. Conclusions

This is the first study to explore the evolution of the research area associated with digital mental well-being promotion and the substantial effect on knowledge production that has occurred since the beginning of the COVID-19 pandemic. It is evident that the pandemic accelerated the pace of research in mental health prevention, with well-being becoming the eighth most frequently reported research theme and mindfulness ranking first in authors’ keywords. Randomised controlled trials are still driving research development. The results indicate that RCTs of digital interventions delivering acceptance and commitment therapy, often to support the well-being of patient populations, have increased the outputs and importance of that intervention approach. At the same time, increased research activity in formative research, with user engagement and usability being classified as niche themes and qualitative research appearing to be a growing area of research, appears to be associated with an increase in interest in a greater range of outcomes within intervention research.

## Figures and Tables

**Figure 1 ijerph-21-00375-f001:**
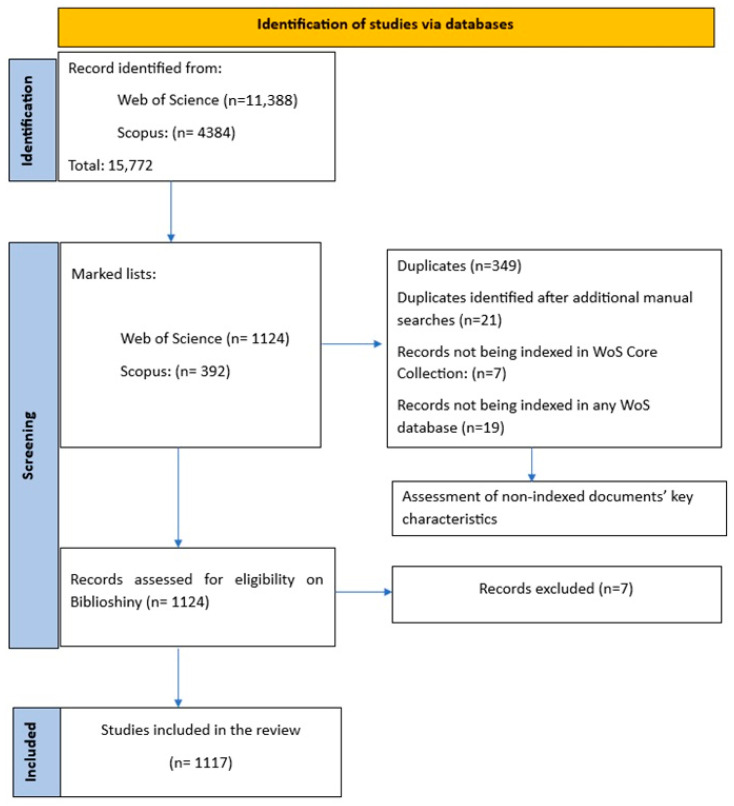
Screening strategy for included documents.

**Figure 2 ijerph-21-00375-f002:**
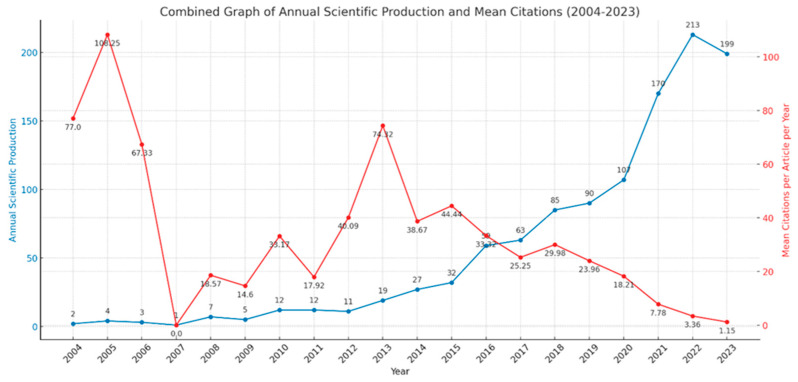
Scientific production rate between 2004–2023.

**Figure 3 ijerph-21-00375-f003:**
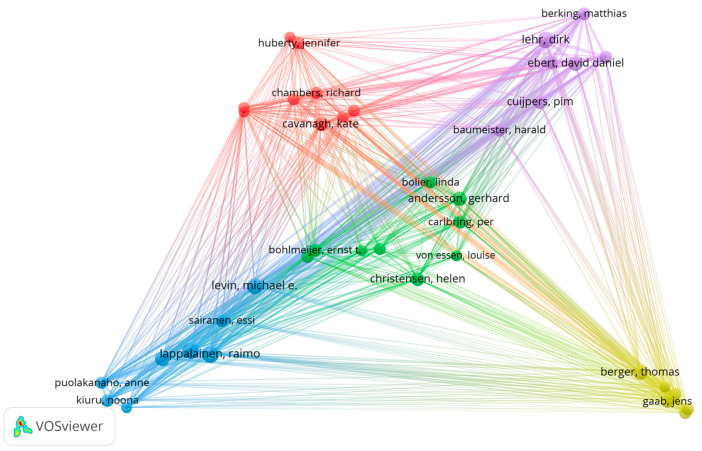
Bibliographical coupling of the most relevant authors.

**Figure 4 ijerph-21-00375-f004:**
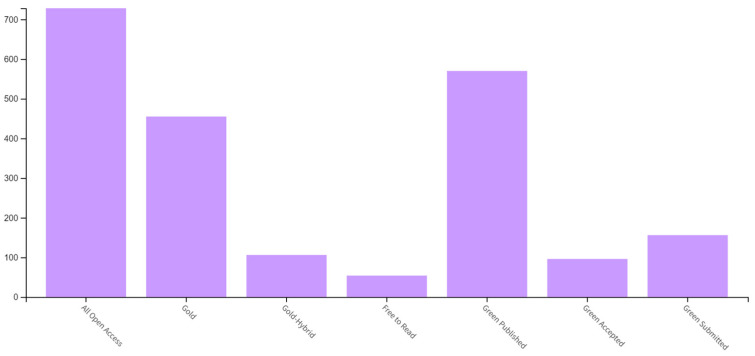
Types of open-access publications.

**Figure 5 ijerph-21-00375-f005:**
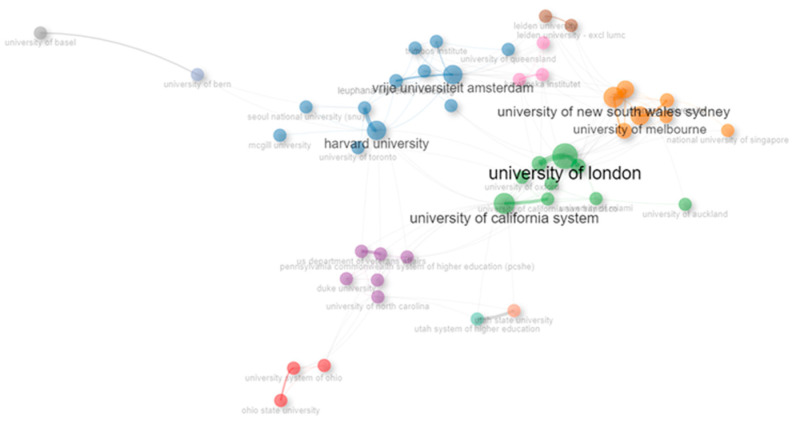
Collaboration networks of correspondent authors’ affiliations.

**Figure 6 ijerph-21-00375-f006:**
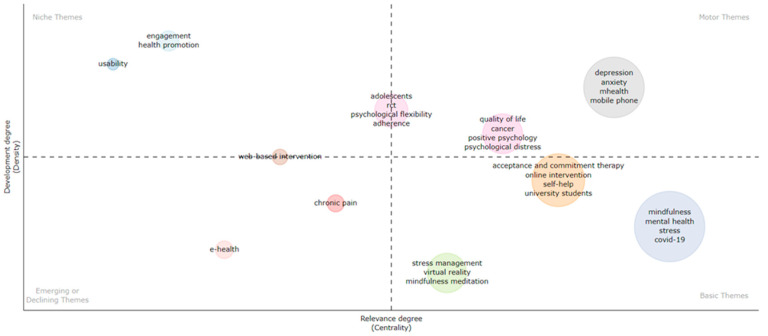
Thematic mapping of main research themes.

**Figure 7 ijerph-21-00375-f007:**
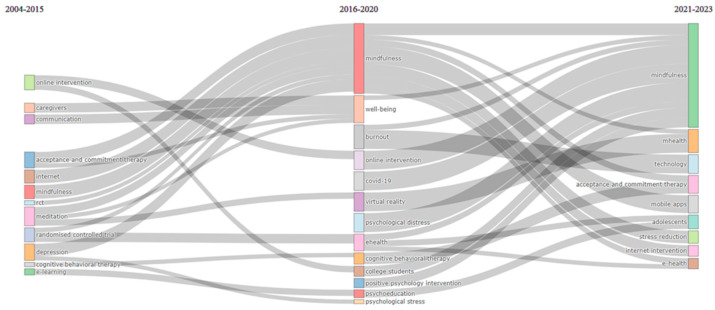
Concept evolution.

**Figure 8 ijerph-21-00375-f008:**
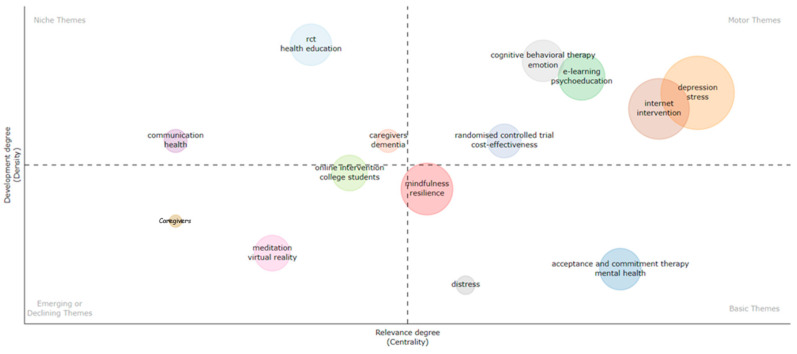
Concept mapping for 2004–2015.

**Figure 9 ijerph-21-00375-f009:**
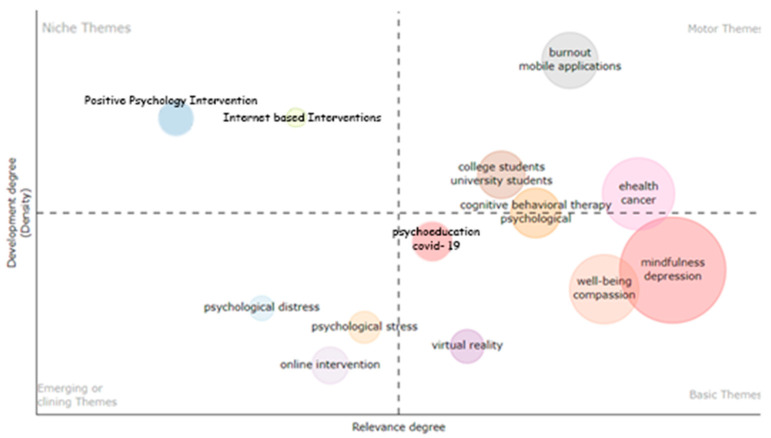
Concept mapping for 2016–2020.

**Figure 10 ijerph-21-00375-f010:**
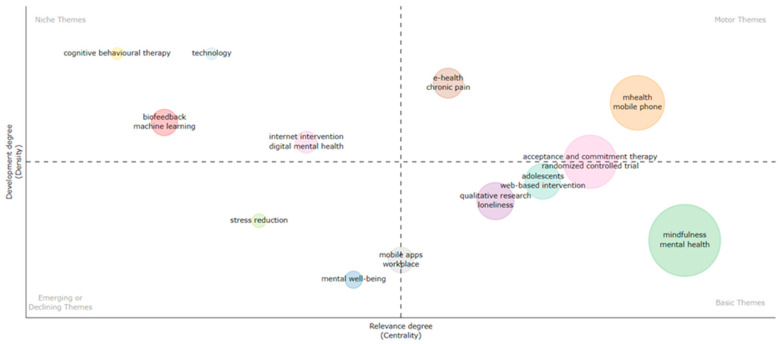
Concept mapping for 2021–2023.

**Figure 11 ijerph-21-00375-f011:**
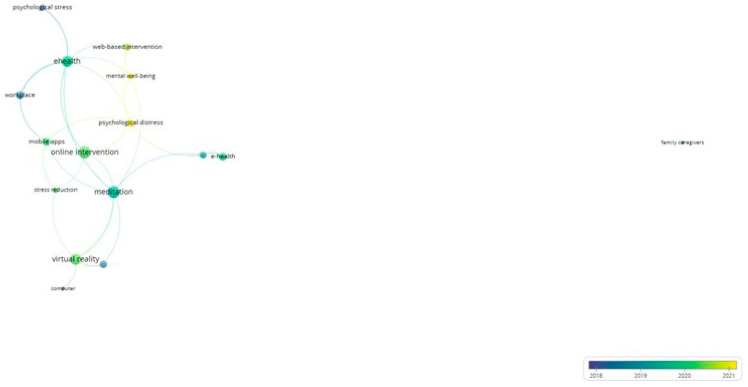
Network analysis of emergent themes.

**Figure 12 ijerph-21-00375-f012:**
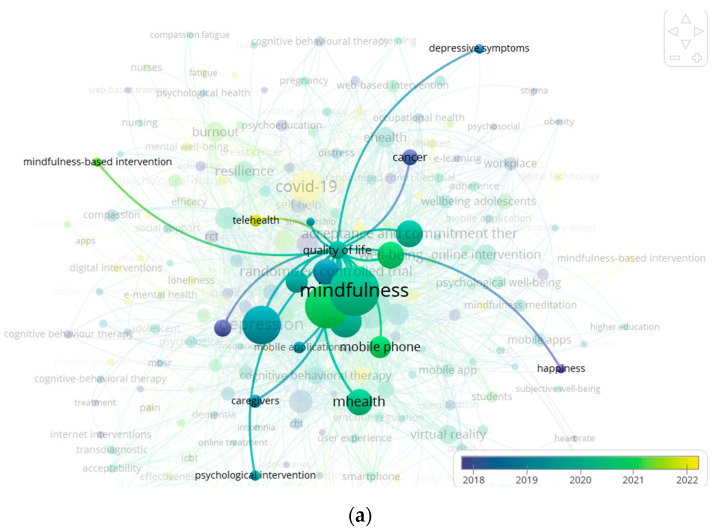
Keywords’ associations for most recently studied psychological/psychosocial outcomes: (**a**) quality of life; (**b**) psychological distress; (**c**) loneliness; (**d**) mental well-being; (**e**) psychological well-being; (**f**) well-being.

**Table 1 ijerph-21-00375-t001:** Most productive authors in digital mental health research: productivity and impact metrics.

Authors	Articles	Articles Fractionalised	h-Index	TC	PY Start	Institution
Lappalainen Raimo	17	2.71	10	482	2013	University of Jyväskylä
Lappalainen Päivi	15	2.55	8	380	2014	University of Jyväskylä
Lehr Dirk	15	2.33	9	509	2014	Leuphana University
Ebert David Daniel	14	2.02	8	496	2014	Technical University Munich
Berger Thomas	13	1.86	6	136	2014	University of Bern
Levin Michael E.	13	3.13	11	452	2014	Utah State University
Kirsikka Kaipainen	11	1.85	6	185	2013	Tampere University

TC: total citation count.

**Table 2 ijerph-21-00375-t002:** Most relevant journals (journals ranking based on Bradford’s law: Zone 1 articles).

Sources	Rank	Articles	h-Index	g-Index	m-Index	TC	PY Start
*Journal of Medical Internet Research*	1	62	22	42	1.10	1864	2005
*JMIR Formative Research*	2	50	6	8	1.20	131	2020
*JMIR Mental Health*	3	43	20	29	2.22	891	2016
*Mindfulness*	4	37	15	26	1.36	701	2014
*Frontiers in Psychology*	5	36	10	19	1.11	401	2016
*JMIR mHealth and uHealth*	6	28	17	28	1.41	1054	2013
*International Journal of Environmental Research and Public Health*	7	27	10	22	1.66	524	2019
*Internet Interventions-The Application of IT in Mental and Behavioural Health*	8	23	11	15	1.37	244	2017
*Annals of Behavioral Medicine*	9	18	3	14	0.20	199	2010
*Psycho-Oncology*	10	17	5	12	0.45	149	2014
*Journal of Contextual Behavioral Science*	11	13	8	10	1.00	117	2017
*Behaviour Research and Therapy*	12	10	9	10	0.60	771	2010
*Frontiers in Psychiatry*	13	10	4	5	0.80	36	2010

**Table 3 ijerph-21-00375-t003:** Top 10 most cited papers.

Documents	Journals	Titles	TC
Blake et al. (2020) [28]	*International Journal of Environmental Research and Public Health*	Mitigating the Psychological Impact of COVID-19 on Healthcare Workers: A Digital Learning Package	321
Gilbody et al. (2015) [29]	*BMJ*	Computerised cognitive behaviour therapy (cCBT) as treatment for depression in primary care (REEACT trial): large scale pragmatic randomised controlled trial	284
Cavanagh et al. (2013) [30]	*Behaviour Research and Therapy*	A randomised controlled trial of a brief online mindfulness-based intervention	283
Clarke et al. (2005) [31]	*Journal of Medical Internet Research*	Overcoming Depression on the Internet (ODIN) (2): A Randomized Trial of a Self-Help Depression Skills Program with Reminders	269
Inkster et al. (2018) [32]	*JMIR Mhealth Uhealth*	An Empathy-Driven, Conversational Artificial Intelligence Agent (Wysa) for Digital Mental Well-Being: Real-World Data Evaluation Mixed-Methods Study	241
Howells et al. (2016) [33]	*Journal of Happiness Studies*	Putting the ‘app’ in Happiness: A Randomised Controlled Trial of a Smartphone-Based Mindfulness Intervention to Enhance Wellbeing	208
Espie et al. (2019) [34]	*JAMA Psychiatry*	Effect of Digital Cognitive Behavioral Therapy for Insomnia on Health, Psychological Well-being, and Sleep-Related Quality of Life: A Randomized Clinical Trial	202
Morris et al. (2010) [35]	*Journal of Medical Internet Research*	Mobile Therapy: Case Study Evaluations of a Cell Phone Application for Emotional Self-Awareness	201
Proudfoot et al. (2013) [36]	*BMC Psychiatry*	Impact of a mobile phone and web program on symptom and functional outcomes for people with mild-to-moderate depression, anxiety and stress: a randomised controlled trial	182
Bostock et al. (2019) [37]	*Journal of Occupational Health Psychology*	Mindfulness on-the-go: Effects of a mindfulness meditation app on work stress and well-being	173

**Table 4 ijerph-21-00375-t004:** Thematic clusters based on authors’ keywords co-occurrence network.

Cluster 1	N	Cluster 2	N	Cluster 3	N	Cluster 4	N	Cluster 5	N	Cluster 6	N	Cluster 7	N
Mindfulness	220	Anxiety	90	COVID-19	102	Acceptance and Commitment Therapy	61	Depression	126	Web-based	24	Positive Psychology	19
Mental Health	168	mhealth	58	Resilience	44	Randomised controlled trial	57	Quality of life	27	College students	19		
Stress	146	Mobile phone	41	Intervention	40	Internet	44	Prevention	20				
Well-being	60	Cognitive behavioural therapy	30	Burnout	36	Internet-based intervention	20	E-health	18				
Stress management	52	Digital health	25	pandemic	18	cancer	20	Qualitative Research	18				
Online	44	Mobile health	24	wellbeing	18	rct	17						
Meditation	43	Mobile apps	19	Psychological wellbeing	17	Chronic pain	17						
Online intervention	41	workplace	19										
Self-compassion	39	Mobile app	18										
Virtual reality	36	ehealth	18										
University students	27	app	17										
Self-help	27	Psychological distress	17										
Adolescents	23	feasibility	16										
		Digital intervention	16										

## Data Availability

The original contributions presented in the study are included in the article; further inquiries can be directed to the corresponding author.

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
