# Peer review of "Evolution of Primary Research Studies in Digital Interventions for Mental Well-Being Promotion from 2004 to 2023: A Bibliometric Analysis of Studies on the Web of Science"

_ijerph, 2024, doi:10.3390/ijerph21030375_

Round 1

Reviewer 1 Report

Comments and Suggestions for Authors

First, thank you for the opportunity to review this work.

The article is interesting and presents a good theoretical and structural framework. It describes a bibliometric analysis of the evolution of research activity from 2004 to 2023, focusing on publication trends of primary studies, authorship, and thematic developments. Bibliometric data were collected on a large sample.

The main limitation of the paper is that the investigation was limited to only two search engines (WoS and Scopus). The article needs to be redefined based on other search engines that can be discussed in parallel (e.g., PubMed).

Although this is described within the limits of the work, I suggest you implement and improve it by comparing other search engines, given the many differences between them, and trying to make a clear analysis of the literature.

Additionally, the implications are poor and should be further argued; it would be necessary to understand better what contribution the following work makes to the scientific community.

The manuscript needs to be better grounded to be published.

Author Response

-The main limitation of the paper is that the investigation was limited to only two search engines (WoS and Scopus). The article needs to be redefined based on other search engines that can be discussed in parallel (e.g., PubMed). Although this is described within the limits of the work, I suggest you implement and improve it by comparing other search engines, given the many differences between them, and trying to make a clear analysis of the literature.

Within this study Scopus searches were compared with searches in the WoS to identify specific limits that exist within WoS searches and understand study characteristics that may be indexed in Scopus and not on WoS. We selected those two databases as Scopus has greater breadth, whereas WoS greater volume of studies.

We agree that there are significant differences across databases which although can provide a more accurate description of the studies’ production it would make it difficult to provide overview of the evolution of research trends and the co-citation patterns. For example, a bibliographical coupling of the most relevant authors was added in the analysis in lines which can only be conducted within one database as long as it allow a full extraction of the bibliometric data of their references. 

For this reason the title of the paper was altered to indicate the scope of the study

-Additionally, the implications are poor and should be further argued; it would be necessary to understand better what contribution the following work makes to the scientific community.

Additional sentences were added in lines 831-837.

“Future research will need to specifically target to review research outputs produced by authors in low- and middle-income countries which would require inclusion of searches in Google Scholar and grey literature. What is more, scoping review techniques can be implemented to conduct searches focusing on specific criteria (e.g. sociodemographic characteristics, health conditions and contexts). Such an approach can allow a better understanding of the impact of the observed research trends in digital interventions.  “

Reviewer 2 Report

Comments and Suggestions for Authors

This is a review of a manuscript titled “Scientific knowledge production in digital interventions for mental well-being promotion: A bibliometric analysis of primary research studies” in which the authors present an analysis of data of research activity of digital interventions. The article is well written; therefore, I suggest the acceptance with minor corrections.  Although, I have some comments and questions that are presented below.

I suggest including in the title the information about knowledge contributors: WoS for studies published between 2004-2023.

I understand the logic behind the analysis, but the objectives of mental well-being promotion are not really considered, observing the differences between journals characteristics.   Have you consider limiting the searches by age, groups, health conditions?

Do you have questioning about why some authors of specific institutions are the ones that have more citations, I mean that what are the specific content of their work. There are any relevant Latin American or African authors which can give information about the general scope of your project. I would like to find more details about a social impact of the results.

Likewise, I think you are missing the characteristics of the journals considering the audiences, if they are open access, the fee to pay. Those conditions are relevant for understanding the editorial approach because if the fee can’t be waived in certain form, it can reflect an obstacle to other authors to have the money to send their work.

Title. Please consider the characteristics of the scientific production in different countries, due to a possible bias about the impact of their research worldwide. Otherwise, I suggest including that most of the work are presented in high-income countries.

Line 29. I suggest including what CDMs you are referring to.

Line 48. Would you consider that low-income countries are different for the application of certain interventions?

Line 109. I recommend mentioning some of the details of this paragraph in the abstract.

Discussion. Although the objective of the manuscript was presenting a description about primary research studies, in the discussion you present elements that try to give information about the interventions, but as I mention before, the characteristics of the journals are important, and the real effect of the interventions could not be a part of the results and interpretation.

Author Response

-I suggest including in the title the information about knowledge contributors: WoS for studies published between 2004-2023.

Information added in the title.

-I understand the logic behind the analysis, but the objectives of mental well-being promotion are not really considered, observing the differences between journals characteristics.   Have you consider limiting the searches by age, groups, health conditions?

Thank you for this suggestion. The objectives of mental well-being promotion as well as the effects of the intervention were not included as that would be beyond the scope of a bibliometric analysis, however we consider such a step to be useful for a scoping and that was added in the future research implications.

“Future research will need to specifically target to review research outputs produced by authors in low- and middle-income countries which would require inclusion of searches in Google Scholar and grey literature. What is more, scoping review techniques can be implemented to conduct searches focusing on specific criteria (e.g. socio-demographic characteristics, health conditions and contexts). Such an approach can allow a better under-standing of the impact of the observed research trends in digital interventions.”

-Do you have questioning about why some authors of specific institutions are the ones that have more citations, I mean that what are the specific content of their work. There are any relevant Latin American or African authors which can give information about the general scope of your project. I would like to find more details about a social impact of the results.

We looked into the authors’ collaboration map that showed that the most relevant authors were the main collaborators in different clusters of authors. As those clusters tended to focus in collaborations among authors within the same countries, we chose to look on institutions’ collaboration maps to understand better the research impact and limitations of those collaborations. A figure was added that showed the bibliographical coupling of the most relevant authors which refers to the presence of authors’ names in the citations of other documents. Also in the lines 262-274 we mention some key research interests of the most prolific authors. The centrality of those interests in the overall research production is then examined via the processes of concept mapping and evolution in later sections.

-Likewise, I think you are missing the characteristics of the journals considering the audiences, if they are open access, the fee to pay. Those conditions are relevant for understanding the editorial approach because if the fee can’t be waived in certain form, it can reflect an obstacle to other authors to have the money to send their work.

We have added a figure showing that the vast majority of the articles were submitted in open-access journals. Although there were no limitations on language and geography the results were dominated by studies from authors affiliated with institutions that could finance these publications.

We have also added information regarding the open-access fee for the top 10 journals in lines 353-360

" As figure 4 shows most of the documents were published in Open Access journals. Among the top 13 most relevant sources, only 5 journals (Mindfulness, Annals of Behavioral Medicine, Psycho-oncology, Journal of Contextual Behavioral Science, and Behaviour Research and Therapy) offered both an Open access and subscription options. Publication fees for open access publications in the top 13 most relevant sources were over $2000 ranging up to $4940.”

-Title. Please consider the characteristics of the scientific production in different countries, due to a possible bias about the impact of their research worldwide. Otherwise, I suggest including that most of the work are presented in high-income countries.

The following line was added in lines 252-253   and further discussed in the discussion :

“Overall, most of the documents were in English language by authors affiliated with institutions in high-income countries”

Line 29. I suggest including what CDMs you are referring to.

Line 48. Would you consider that low-income countries are different for the application of certain interventions?

 Yes because factors that influence intervention implementation and acceptability can be expected to differ. Unfortunately, studies form low-income countries were significantly under-represented in sample studies. We consider that the study of digital interventions across those countries would require a search strategy that would specifically target document sources and authors that report them otherwise insights will reflect findings in different contexts. The fact that they have such a limited representation in the Web of Science proves this.

We have added the following statements:

Lines 677-683:

“Overall, most of the work focused was produced by authors with institutions affiliations in high-income countries with open-access publications driving research publications. This means that studies’ insights may be less relevant to resource-poor settings, especially low-and-middle-income countries.

 lines 804-807:

“An associated limitation of this study was that the sample of documents was dominated by studies conducted in high-income countries despite the absence of language re-strictions in our searches”

Lines 831-837:

“Future research will need to specifically target to review research outputs produced by authors in low- and middle-income countries which would require inclusion of searches in Google Scholar and grey literature. What is more, scoping review techniques can be implemented to conduct searches focusing on specific criteria (e.g. sociodemographic characteristics, health conditions and contexts). Such an approach can allow a better understanding of the impact of the observed research trends in digital interventions. 

Line 109. I recommend mentioning some of the details of this paragraph in the abstract.

-Abstract updated

Discussion. Although the objective of the manuscript was presenting a description about primary research studies, in the discussion you present elements that try to give information about the interventions, but as I mention before, the characteristics of the journals are important, and the real effect of the interventions could not be a part of the results and interpretation.

We have added a paragraph in the discussion focusing on the open-access publications and its impact in scientific knowledge production. The structure of the second part of the discussion aims to interpret the findings in the diagrams that explain the evolution and changes in research concepts as defined by the keywords that authors used to index their studies. As a result, it aims to provide adequate information on the research landscape where those differentiations appear.

Round 2

Reviewer 1 Report

Comments and Suggestions for Authors

Dear Authors, 

I appreciate your efforts in revising your article. 

In reading it, I found the highlighted sections implemented and clarified. Despite the discussed limitations, I read the work more thoroughly and clearly. 

Best regards